Two new species of Primulina (Gesneriaceae) from limestone karsts of China

Hong Xin 1 2
Li Zhong-Lin 3
Liu Jia-Zhi 1
Zhou Shou-Biao 2
Qin Wei-Hua 95235961@qq.com 3
Wen Fang wenfang760608@139.com 4
1 Anhui University , Hefei , Anhui , China
2 Anhui Normal University , Wuhu , Anhui , China
3 Nanjing Institute of Environmental Sciences, Ministry of Ecology and Environment of the People’s Republic of China , Nanjing , Jiangsu , China
4 Guangxi Key Laboratory of Plant Conservation and Restoration Ecology in Karst Terrain, Guangxi Institute of Botany , Guiling , Guangxi , China
Giordani Paolo
Electronic publication date: 2018 Jun 11
Publication date: 2018
Volume: 6
Electronic Location ID: e4946
Received 2018 Feb 21; Accepted 2018 May 21
Copyright: ©2018 Hong et al.
Copyright year: 2018
Copyright holder: Hong et al.
License: This is an open access article distributed under the terms of the Creative Commons Attribution License, which permits unrestricted use, distribution, reproduction and adaptation in any medium and for any purpose provided that it is properly attributed. For attribution, the original author(s), title, publication source (PeerJ) and either DOI or URL of the article must be cited.
License URL: https://creativecommons.org/licenses/by/4.0/

Keywords: Karst region, Didymocarpinae, Taxonomy, Morphology, Primulina davidioides, Primulina hiemalis

Funding: Anhui University Doctor Startup Fund Key University Science Research Project of Anhui Province KJ2017A022 Fund of Guangxi Key Laboratory of Plant Conservation and Restoration Ecology in Karst Terrain 16-B-01-01 Plant germplasm resources projects of the germplasm bank of Wild species of Kunming Institute of Botany Chinese Academy of Sciences WGB-1411 Chinese Academy of Sciences under the Guangxi Natural Science Foundation 2015GXNSFBB139004 Chinese Academy of Sciences under the STS initiative “Development of Chinese Union of Botanical Gardens” KFJ-1W-NO1 This work was supported by Anhui University Doctor Startup Fund, Key University Science Research Project of Anhui Province (No. KJ2017A022), Fund of Guangxi Key Laboratory of Plant Conservation and Restoration Ecology in Karst Terrain (16-B-01-01), Plant germplasm resources projects of the germplasm bank of Wild species of Kunming Institute of Botany, Chinese Academy of Sciences (WGB-1411), the Chinese Academy of Sciences under the Guangxi Natural Science Foundation (2015GXNSFBB139004) and the Chinese Academy of Sciences under the STS initiative “Development of Chinese Union of Botanical Gardens” (KFJ-1W-NO1). The Gesneriads Society, The African Violet Society of America helped contact the botanical collections to facilitate our access to examined specimens, and hosted the first author when he did research in the US and Canada.

==============================
The limestone karst area of South China is a major biodiversity hotspot of global terrestrial biomes. During extensive field work on the Guangxi limestone formations, two unknown species of Gesneriaceae were collected. After conducting a comprehensive study of the literature and herbarium specimens, Primulina davidioides and P. hiemalis are recognized as two species new to science, and described and illustrated here. P. davidioides is morphologically close to P. lunglinensis based on the shape of the leaf and flower, but it can be easily distinguished by the shape of the bracts, corolla and stigma, indumentum of peduncles, pedicels and pistil and number of staminodes. P. hiemalis is closely relate to P. luzhaiensis in vegetative appearance, but differs in the shape of the calyx and stigma, number of bracts and staminodes, indumentum of the leaf blade and peduncle, and position of stamens in the corolla tube. Considering that not enough is known about their populations, it is proposed that their conservation statuses should currently be classed as data deficient (DD) according to the IUCN Red List Category and Criteria.

Introduction

The tropical and subtropical karst landforms of southern and southwestern China are renowned because of their unrivalled biodiversity and high endemism among the tropical and subtropical floras of the world (Myers et al., 2000; Clements et al., 2006; Hou et al., 2010). Karst areas in South China offer a multitude of ecological niches for plant diversification and speciation (Ai et al., 2015), with about half of all the endemic genera of flowering plants in China (Ying & Zhang, 1994). Among these, the Gesneriaceae form the most abundant with 28 genera amounting to 90% of all endemic genera of the family in China (Wang, Pan & Li, 1990; Wang et al., 1998; Li & Wang, 2004; Möller et al., 2016). At the same time, China is a significant centre of diversity of Old World Gesneriaceae, which consists of 52 genera (Möller et al., 2016), with 75% of all species endemic to this region (Gao et al., 2015).

A great number of new species were described in the genus Primulina (Gesneriaceae) in recent years (Yang et al., 2018), and it is becoming one of the most interesting genera of the Old World Gesneriaceae, comprising more than 170 species (Wang et al., 2011; Möller et al., 2011; Möller et al., 2016). This group shows high levels of endemism and ecological specialisation (Gao et al., 2015), with narrow island distributions (Wang et al., 1998; Li & Wang, 2004; Wei et al., 2010), i.e., only in karst towers and caves (Ai et al., 2015). The limestone regions of southern and southwestern China and northern Vietnam possess the highest biodiversity of Primulina with about 80% of species endemic here (Wei et al., 2010). Many Primulina species pairs can successfully interbreed through artificial experiments (Wen, 2008; Zhang, Yang & Kang, 2017), suggesting that Primulina is probably a genus under recent or ongoing speciation and differentiation (Gao et al., 2015).

During our continuous floristic surveys of limestone karsts flora in 2009 and 2010, we revealed an additional two species of Primulina not previously known. Further fieldwork was conducted at the same locality, and flowering specimens were collected. Available information suggested that these two species are rare and usually occur only at one or two localities. After a comprehensive analysis of the literature (Wang, Pan & Li, 1990; Wang et al., 1998; Li & Wang, 2004; Wei et al., 2010), as well as herbarium specimens of E, GH, HITBC, HN, IBK, IBSC, K, MO, KUN, PE and US (herbarium acronyms according to Index Herbariorum; Thiers, 2017), the specimens were identified as two new taxa of Primulina based on results of detailed examination of morphological anatomical features, which we hereby describe and illustrate.

Materials and Methods

Ethics statement

All the collecting locations of the new species reported in this study are not in any natural conservation area and no specific permissions were required for these locations. Since the species are currently undescribed, inevitably, they are not currently included in the China Species Red List (Wang & Xie, 2004). Our field studies did not involve any endangered or protected species. No specific permits were required for the present study.

Nomenclature

The electronic version of this article in Portable Document Format (PDF) will represent a published work according to the International Code of Nomenclature for algae, fungi, and plants (ICN), hence the new names contained in the electronic version are effectively published under that Code from the electronic edition alone. In addition, new names contained in this work which have been issued with identifiers by IPNI will eventually be made available to the Global Names Index. The IPNI LSIDs can be resolved and the associated information viewed through any standard web browser by appending the LSID contained in this publication to the prefix “http://ipni.org/”. The online version of this work is archived and available from the following digital repositories: PeerJ, PubMed Central, and CLOCKSS.

Material collection

These two species were collected and examined during the floristic field surveys. In addition, the plants were monitored in the field and nursery in the Gesneriad Conservation Center of China (GCCC) by the authors over the past eight years. We collected leaf materials of these possible new species, using silica gel to dry them in the field for DNA extraction. We also cultivated about ten young plants from leaf cuttings of each new species for ex situ conservation in the Gesneriad Conservation Center of China (GCCC) so that we can preserve the germ plasma resource of this rare species.

Morphological observations and specimens examined

An overview of the genus Primulina from southern China and adjacent areas of northern Vietnam was prepared. All available specimens of Primulina stored in the following herbaria were carefully examined: E, GH, HITBC, HN, IBK, IBSC, K, MO, KUN, PE and US. The images of type specimens were also obtained from Tropicos (http://www.tropicos.org) and JSTOR Global Plants (http://plants.jstor.org). Morphological description of the new species was based on examination of fresh and pressed specimens. All morphological characters, particularly the details of the flowers, were studied under a stereomicroscope (MD-90, Olympus, Tokyo, Japan), and are described using the terminology presented by Wang et al. (1998). The morphological comparison with other species was based on study of live plants in the field and in cultivation in GCCC, herbarium specimens, and also information gathered in the literature searches.

Results

Primulina davidioides F. Wen & Xin Hong sp. nov. (Figs. 1 and 2)

Figure 1 Illustration of Primulina davidioides sp. nov.

(A) habit in flowering period; (B) dissection of a flower showing stamens and staminodes; (C) anthers, showing beard; (D) pistil; (E) stigma; (F) ovary and calyx; (G) calyx (Image credit: Xiao-Ming Xu, drawn from the holotype).

Figure 2 Photographs of Primulina davidioides sp. nov.

(A) habitat; (B) population; (C) the flower bud; (D) flowering plant with cymes; E–H: cymes: (E) showing adaxial bracts; (F) showing abaxial bracts; (G) frontal view of cymes; (H) top view of cymes); (I) lateral view of corolla; (J) young fruit; (K) top view of corolla; (L) upward view of corolla; (N) opened corolla, pistils without corolla and calyx lobes; (O) dissection of a flower showing staminodes; P–Q: stamens (P) the reverse side, showing beard; (Q) the frontal side, showing the anthers; (R) capsules (Photo credit: Fang Wen and Xin Hong).

IPNI:

Type. China. Guangxi: Donglan County, Donglan Town, Dayou village, growing in rocky crevices and tufa surface on the bottom of a limestone hill, elevation ca. 350 m, 05 May 2012 (fl.), F. Wen & Xin Hong 201205005 (holotype: IBK; isotype: AHU).

Paratypes. China: Guangxi: Donglan County, Donglan Town, Dawen Village, ca. 350 m, 12 May 2009, F. Wen 0905012 (IBK); ibid., 350 m, 28 May 2011, Xin Hong 201209030 (AHU).

Diagnosis. P. davidioides is morphologically close to P. lunglinensis based on elliptical to broadly ovate leaf blade, purple corolla and cylindrical tube, but it can be easily distinguished by puberulent peduncles (vs. glandular puberulent); bracts cordate to suborbicular, base subtruncate, 4–6 × 4–5 cm (vs. elliptic to broadly ovate, base attenuate, 0.6–3.8 × 0.4–2.7 cm); pedicels glandular and eglandular–pubescent (vs. glandular puberulent); corolla ca. 6 cm long (vs. 3–3.8 cm long), staminodes 3 (vs. 2), pistil ca. 3.7 cm long, glandular and eglandular pubescent (vs. ca. 2.7 cm long, puberulent), stigma narrowly obtrapeziform (vs. cuneate) (see Table 1).

Table 1 Diagnostic character differences between Primulina davidioides sp. nov. and its close relatives P. lunglinensis.

Characters	P. davidioides	P. lunglinensis	
Indumentum of peduncles	puberulent	glandular puberulent	
Bracts	cordate to suborbicular, base subtruncate, 4–6 × 4–5 cm	elliptic to broadly ovate, base attenuate, 0.6–3.8 × 0.4–2.7 cm	
Indumentum of Pedicel	glandular and eglandular pubescent	glandular puberulent	
Corolla size	ca. 6 cm long	3–3.8 cm long	
Number of staminodes	3	2	
Pistil	ca. 3.7 cm long, glandular and eglandular pubescent	ca. 2.7 cm long, puberulent	
Shape of Stigma	narrowly obtrapeziform	cuneate	

Description. Perennial. Rhizome terete, 2–6 cm long, 1–2 cm in diameter. Leaves in basal rosette, 4–10, opposite; pachyphyllous, rigid and coriaceous when dry, densely pubescent on both surfaces; petiole flattened, 1–2 cm long, 6–10 mm wide, densely pubescent; blade elliptical to broadly ovate, (5–) 11–12.5 cm, 5–10 cm wide, apex somewhat obtuse, base obliquely cuneate, margin shallowly serrate; lateral veins 3–5 on each side, impressed adaxially and prominent abaxially. Cymes 1–3, 5–9 (11)-flowered; peduncles 5–9 (12) cm, 4–8 mm in diameter, pubescent; pedicels ca. 1.4 cm long, glandular and eglandular-pubescent; bracts 2, opposite, free, cordate to suborbicular, thickly chartaceous, slightly fleshy, leathery when dry, white when flowering, with crinkled and virescent margins, 4–6 cm long, 4–5 cm wide, base subtruncate, apex acute, outside densely puberulent and sparsely strigillose, inner surface glandular-puberulous. Calyx membranous, white to subtranslucent, ca. 7 mm, 5-sect; segments equal, triangular, 2–3 mm wide, margin obscurely serrated from the middle, apex acute, outside densely puberulent. Corolla ca. 6 cm long, purple, with dark purple lines inside, outside glandular and eglandular–pubescent, inside pubescent near base; tube cylindrical; ca. 4.5 cm long, ca. 1 cm diameter at mouth, ca. 0.5 cm in diameter at base; limb distinctly 2-lipped, adaxial lip 2-parted to the base, lobes ovate, ca. 6 × 5 mm, 3-lobed from near the middle, lobes oblong, 10–15 × 4–6 mm. Stamens 2, abaxial, adnate to ca. 1.7 cm above corolla base; filaments white, ca. 1.1 cm, strongly geniculate near the base, ca. 1 mm wide, sparsely glandular–puberulous; anthers fused along their entire adaxial surfaces, reniform–oblong, ca. 3 mm long, ca. 2 mm wide, pale yellow, bearded on the back. Staminodes 3, adaxial, linear, apex capitellate, lateral ones ca. 7 mm long, adnate to ca. 1.5 cm above corolla base, central one ca. 4.5 mm long, adnate to ca. 0.9 cm above corolla base, sparsely glandular-puberulous. Disc ring-shaped, indistinctly lobed, 0.5–0.8 mm high, glabrous. Pistil linear, ca. 3.7 cm long, green, densely puberulent with both glandular and eglandular hairs; ovary ca. 2.0 cm long, ca. 1.5 mm wide. Lower lip of stigma narrowly obtrapeziform, apex retuse, translucent to green, ca. 3 mm long, recurved. Capsule ca. 5 cm long, brownish.

Distribution, Habitat and Ecology. Primulina davidioides is rare, only found in the type locality, i.e., an unnamed limestone hill ca. 20 km west of Donglan Town, Donglan County, in northern Guangxi province of southern China. It grows in rocky crevices and tufa surface on the bottom of a limestone hill at an elevation of 350 m a.s.l. The average temperature of Donglan County is 18.7 °C, the average annual precipitation has been calculated at ca. 1,660 mm. The forest where P. davidioides occurs is subtropical evergreen broad-leaf forest. Flowering in May, fruiting in September.

Etymology. The specific epithet is derived from its cordate to suborbicular bracts with subtruncate base. The shape of the bracts looks similar to the bracts of Davidia involucrate Baill., native to China.

Primulina hiemalis Xin Hong & F. Wen sp. nov. (Figs. 3 and 4)

Type: China, Guangxi, Yongfu county, Baishou Town, Chuanyan village, on the moist rock face at the entrance of limestone caves, 526 m a.s.l., 09 December 2010, F. Wen & L.F. Fu WFBCJT101209-01 (holotype: IBK; isotype: AHU).

Additional collections. China, Guangxi, Yongfu county, Baishou Town, Baishouyan, on the moist rock face at the bottom of limestone hills, 511 m a.s.l., 15 December 2011, F. Wen WFBCJT111215-01 (IBK).

Figure 3 Illustration of Primulina hiemalis sp. nov.

(A) habit in flowering period; (B) dissection of a flower showing stamens and staminodes; (C) calyx and pistil; (D) calyx, showing puberulence (Image credit: Wen Ma, drawn from the holotype).

Figure 4 Photographs of Primulina hiemalis sp. nov.

(A) habitat; (B) plant; C–F: corolla: (C) frontal view; (D) right side view; (E) top view; (F) left side view; (G) calyx; (H) pistils with calyx lobes, showing the stigma; (I) opened corolla (Photo credit: Fang Wen).

Diagnosis. Morphologically, Primulina hiemalis resembles P. luzhaiensis since both species having thick chartaceous leaves, obliquely ovate or oblong leaf blade, purple corolla, 4–6 cm, tubular tube. But it can be readily distinguished from P. luzhaiensis by the indumentum of the leaf blade and peduncle (puberulent vs. villous); bracts 3 (vs. 2), lateral ones ca. 2 cm long (vs. 0.2–0.8 mm long); longer calyx, ca. 1.5 cm long (vs. 0.5–1 cm long); stamens adnate to ca. 2 cm (vs. 1.4–1.7 mm) above the corolla tube base; staminodes 2 (vs. 3), stigma ligulate, apex 2-lobed to middle (vs. obtrapeziform, apex retuse) (shown in Table 2).

Description. Perennial. Rhizome subterete, ca. 1.0–2.5 × 0.5–2.0 cm wide, internodes inconspicuous. Leaves in basal rosette, 6–8, opposite; petiole 0.5–3.5 × 0.4–0.6 cm; leaf blade thickly chartaceous, markedly obliquely ovate or oblong, 5.5–9.5 × 3.5–5.0 cm, apex obtuse, base obviously oblique, cuneate–attenuate, margin crenate, obviously serrated from the base; densely puberulent on both surfaces, lateral veins 4–5 on wider side of midrib, lateral veins inconspicuous, 3–4 on narrower side of midrib. Cymes 2–4, 1–3-flowered; peduncle 1.5–3.0 cm, densely puberulent; bracts 3, free, narrowly lanceolate, lateral ones ca. 20 × 2–2.5 mm, the central one ca. 5 × 0.8–1 mm, puberulent, margin entire, apex acute. Pedicel 0.5–1.0 cm, densely puberulent. Calyx 5-sect from base; segments equal, lanceolate-ovate, ca. 15 × 6 mm, puberulent, margin entire, apex acute. Corolla purple, 4–6 cm, outside puberulent, inside glabrous; tube tubular, 2.5–4.8 cm long, base ca. 3–5 mm in diameter, top 12–16 mm in diameter; limb distinctly 2-lipped, adaxial lip 2-parted to the base, lobes ovate, ca. 10 × 8 mm, 3-lobed over the middle, lateral ones ovate, central one oblong, obtuse to truncate at apex, 12–15 × 8–10 mm. Stamens 2, adnate to ca. 2 cm above the corolla tube base, filaments ca. 1 cm long, geniculate near the base, glabrous; anthers fused by their entire adaxial surfaces, elliptic, ca. 2.5 mm long, glabrous; staminodes 2, linear, 6–7 mm, glabrous. Pistil ca. 3 cm, densely puberulent with both glandular and eglandular hairs; ovary ca. 2.0 cm, glandular-puberulent. Stigma ligulate, lobes linear, 2–2.5 mm long. Capsule 2.5–3.0 cm long.

Distribution, Habitat and Ecology. Primulina hiemalis is locally abundant, narrowly endemic and only known from the type locality, i.e., Chuanyan village, Baishou Town, Yongfu county, Guangxi province. It grows on moist, shady cliffs at the entrance of a limestone cave, at an elevation of 530 m a.s.l. The average temperature is 18.8 °C, the average annual precipitation has been calculated at ca. 2,000 mm. The forest where P. hiemalis occurs is subtropical evergreen broad–leaf forest. Flowering from November to January, fruiting in April.

Table 2 Diagnostic character differences between Primulina hiemalis sp. nov. and its close relatives P. luzhaiensis.

Characters	P. hiemalis	P. luzhaiensis	
Indumentum of leaf blade	puberulent on both surfaces	villous and pubescent adaxilly	
Indumentum of peduncle	densely puberulent	densely villous	
Bracts	3, lateral ones ca. 2 cm long, the central one ca. 0.5 cm long	2, 0.2–0.8 mm long	
Calyx size	ca. 1.5 cm long	0.5–1 cm long	
The location of stamens	adnate to ca. 2 cm above the corolla tube base	adnate to 1.4–1.7 mm above the corolla tube base	
Number of staminodes	2	3	
Shape of stigma	ligulate, apex 2-lobed to middle	obtrapeziform, apex retuse	

Etymology. The specific epithet is derived from the species’ winter flowering time. The Latin word “hiemalis” is an adjective, meaning “to belong to winter,” and hints at the flowering time of the new taxon (November to January).

Discussion

In contrast to its high species diversity, the morphological variation of Primulina is relatively limited compared to other genera (Möller et al., 2016). The corolla morphology, especially, is relatively uniform, and most species possess straightly infundibuliform corollas, with only the salverform (e.g., Primulina tabacum), campanulate (e.g., P. dichroantha, P. mollifolia, P. hezhouensis, and P. renifolia), and P. curvituba having a strongly curved tube shape. In addition, Primulina displays a wide range of diversity of involucral bracts. The two opposite bracts are brightly white when flowering and serve the function of attracting pollinators as in Primulina eburnea, P. lutea, P. xiziae, P. lungzhouensis, P. lunglinensis, P. beiliuensis var. beiliuensis, P. beiliuensis var. fimbribracteata and the new species, P. davidioides. The development of cymes in these species shows that the two lateral paraclades of the pair-flowered cyme are reduced, so all flowers are clustered together. While flowers of these species always blossom for a short duration (Hong, 2016), the large white bracts and central flower cluster make the inflorescence look superficially like a single flower, perhaps to increase attraction for pollinators.

Conservation Aspects

These two new species belong to a group of stenochoric plants which only grow in limestone areas. Currently there is insufficient information concerning the distribution and population status of these new species. Obviously, further field study is needed in northwestern Guangxi as their geographic range may well be more extensive than presently known. Considering that not enough is known about their populations, it is proposed that their conservation statuses should currently be classed as data deficient (DD) (IUCN, 2016).

Conclusions

During our continuous floristic surveys of limestone karsts in recent years, two unknown species of Gesneriaceae were collected. After conducting a comprehensive study of the literature and herbarium specimens, Primulina davidioides and P. hiemalis are recognized as two species new to science based on results of detailed examination of morphological anatomical features. Current information for these species is only known from very few collections, they appears to be narrowly endemic and locally abundant based on our careful field investigations in the past eight years.

The authors express our appreciation to the corresponding curators of the botanical collections for facilitating our access to examined specimens. We thank Long-Fei Fu and Jia Dong for help in cultivating works and preparing comparative materials; Zheng-Long Li, Fu-Zhuan Pan, Shu Li and Zi-Bing Xin for help conducting fieldwork and with logistics; and Elizabeth Mchone from Drexel University for reviewing the English grammar and style. Thanks are also due to Xiao-Ming Xu and Wen Ma for the beautiful hand-drawings. We also grateful to Michael Möller and two anonymous reviewers for the valuable comments that improved the manuscript.

Additional Information and Declarations

Competing Interests

Author Contributions

Data Availability

New Species Registration

The authors declare there are no competing interests.

Xin Hong conceived and designed the experiments, performed the experiments, analyzed the data, prepared figures and/or tables, authored or reviewed drafts of the paper, approved the final draft.

Zhong-Lin Li and Jia-Zhi Liu conceived and designed the experiments, performed the experiments, contributed reagents/materials/analysis tools, prepared figures and/or tables, authored or reviewed drafts of the paper, approved the final draft.

Shou-Biao Zhou analyzed the data, authored or reviewed drafts of the paper, approved the final draft.

Wei-Hua Qin conceived and designed the experiments, analyzed the data, contributed reagents/materials/analysis tools, authored or reviewed drafts of the paper, approved the final draft.

Fang Wen conceived and designed the experiments, analyzed the data, authored or reviewed drafts of the paper, approved the final draft.

The following information was supplied regarding data availability:

All herbarium specimens, leaf materials and seeds were deposited in both the Anhui University Museum (AHU) and herbarium of Guangxi Institute of Botany (IBK).

The following information was supplied regarding the registration of a newly described species:

Primulina davidioides F. Wen & Xin Hong sp. nov. LSID: 77179004-1; Primulina hiemalis Xin Hong & F. Wen sp. nov. LSID: 77179005-1.

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
