# Peer review of "Two new species of Primulina (Gesneriaceae) from limestone karsts of China"

_PeerJ, doi:10.7717/peerj.4946_

## Round 0.1 · original submission · Major Revisions

In particular, the reviewers and I concur that significant improvements of the English grammar and style are needed. Furthermore, in its present form, the discussion is too long and not fully fitting the presented results.

Reviewer 1 ·

Basic reporting

The article needs significant improvements of the English grammar and style throughout. There were places that citations were barely paraphrased from the source material (Lines 45-48). There are more than enough citations and proper background is given, but it is hard to follow. Table 1 and 2 should be condensed to show the significant differences that one would use to separate the taxa efficiently. It might be more useful to see a dichotomous key to the taxa as opposed to the tables. Figure 1 would be better called "Illustration of Primulina davidioides." Although Figures 3 and 4 are beautiful, I think that again a single figure noting differences would be more effective. The purpose is to quickly notice the differences and be able to compare with other publications or field specimens.

Experimental design

I am okay with the methods there appears to be enough replication and specimens viewed from many sources.

Validity of the findings

I have no doubt that these are two new species. There are obvious differences shown but the overall paper could be much more concise.

Additional comments

You have done an extensive amount of work and I am happy to see two wonderful new species. I hope that you are able to edit the grammar and style.

Reviewer 2 ·

Basic reporting

Figure 1.
In the photograph (Fig. 3) and manuscript, P. davidioides has three staminodes. However, I cannot find the third staminode in Figure 1-B. I recommend you to use another illustration or to add the third staminode in the illustration.

Experimental design

no comment

Validity of the findings

no comment

Additional comments

Dear authors, 
Hong et al. found two new Primulina species in limestone karsts of Southern China, and taxonomic investigations for the plants were conducted based on detailed examination of morphological anatomical features. I consider your manuscript to be well written, and the new Primulina introduced in this study will be good species. In addition, I think that this work may have a good contribution to the understanding of the plant diversity in China. Nevertheless, I found some minor points to add/change. Please see following comments.

Specific comments:

Page 7, L 52. I think that “Fig. 1” cited at the end of the sentence is inadequate and unnecessary. Please delete it.

Page 10, L96. “Japan”

Page 10, L108-109. You should explain the validity of the basis, why one could argue that P. lunglinensis is closely related to P. davidioides. What kind of floral and/or leaf morphology? Please clarify.

Page 12, L157. “vegetative appearance” is ambiguous. Please explain the details of that.

Page 14, L184. “November” or “December”, be consistent.

·

Basic reporting

The English quite poor in grammar and style and requires a substantial input prior publication.
References are sufficient, but need reassessment after final revision of the MS
Structure: The discussion is too long and over most parts irrelevant to the presented data, i.e. the morphological description of two new species. It is unclear what the purpose of the discussion is. Removing the irrelevant or too general parts and moving the family info to introduction, little it left. This is perhaps not the best place to discuss the floral diversity of a genus when only two new species are described and without any hypothesis to test. A discussion here could surround the wider morphological affinities of the new species in the genus. Only the two comparative species are mentioned, but the vegetative habit of P. davidioides is very generic and its occurrence in other species could be discussed, and the floral affinities in a wider genus-wide context, but compared to other specific species, not just in general terms.
Some parts from the discussion are better placed in the introduction to introduce the plant family Gesneriaceae. This is rather absent in the introduction at present.
The conservation assessment should go before the discussion separately under each species with detailed justification for the categories and subcategories chosen.

Self-contained, though the discussion too general and mostly irrelevant.

Experimental design

Ok

Validity of the findings

Further specific comments:
Line 43: Add Möller et al 2016 Guihaia here for an updated account of endemic gesner genera in China.
Line 78: LSIDs are not available for the new species, but are apparently a requirement of PeerJ
Line 85: More details on the nature of material deposited is needed, e.g. herbarium specimens? spirit preserved? what tissue samples were deposited?
Line 93: How were type images consulted in IPNI?

Line 108: The diagnosis needs more focus on fewer main characters and a listing of the respective characteristics for P. lunglinensis. Other characters to consider are given below:
P. lunglinensis P. davidioides
Corolla with two yellow ridges on floor with yellow ridges
Filaments red white
Petiole 0.6-8 cm 1-2 cm
The leaves and large bracts of the new species are very similar to P. lutea. This can be discussed in the notes/discussion.
Lines 126+: These yellow lines are difficult to see in the photos. Are these located on ridges as in P. lunglinensis?
Line 132: describe anthers coherence.
Lines 157+: see comments for diagnosis for P. davidioides re characteristics of P. luzhaiensis.
Line 157: the leaves and their arrangement are very common in Primulina and thus characters of those are not useful.
Line 175: Are they really fused, or coherent by long papillae as in Saintpaulia? See Weberling 1981: Morphologie der Blueten und der Bluetenstaende, Ulmer Verlag, page 131. Please check for both species
Line 195: These numbers are out of date. Check Möller et al 2016 Guihaia
Line 190+: The discussion is mostly not relevant to the two species described here. The beginning seems like a good introduction of the family and genus in China.
Lines 254+: The conservation part is greatly repeat of earlier text and can be reduced and focussed on the two species and their threats, to explain their categorisation.
Lines 294+: The references need adjustment at the end of the revision of the MS.

Additional comments

see above

---

## Round 0.2 · Minor Revisions

Several minor revisions are still requested by the reviewer. Their comments about the English language echo comments from the initial round by other reviewers, so please give this the additional attention which is needed.

·

Basic reporting

English of the newly added text still occasionally to be corrected.
Spelling mistake of Gesneriad (line 86).

Experimental design

ok

Validity of the findings

The conservartion assessments need more details, which will provide important baseline data as future reference point.
However, the proposed categories need furter thoughts and perhaps if uinsufficient knowledge exists, they maybe DD, data deficient. See comments in MS.

Additional comments

Needs some more work as described above and in the annotated MS file.

---

## Round 0.3 · accepted · Accept

I think that, in the revised version, you fulfilled the requests made by the reviewer.

#